# Harnessing the Benefits of Endogenous Hydrogen Sulfide to Reduce Cardiovascular Disease

**DOI:** 10.3390/antiox10030383

**Published:** 2021-03-04

**Authors:** Kevin M. Casin, John W. Calvert

**Affiliations:** Carlyle Fraser Heart Center, Division of Cardiothoracic Surgery, Department of Surgery, Emory University School of Medicine, Atlanta, GA 30322, USA; kcasin@emory.edu

**Keywords:** heart, cardiovascular disease, heart failure, diabetic cardiomyopathy, hydrogen sulfide, metabolism, fasting, exercise

## Abstract

Cardiovascular disease is the leading cause of death in the U.S. While various studies have shown the beneficial impact of exogenous hydrogen sulfide (H_2_S)-releasing drugs, few have demonstrated the influence of endogenous H_2_S production. Modulating the predominant enzymatic sources of H_2_S—cystathionine-β-synthase, cystathionine-γ-lyase, and 3-mercaptopyruvate sulfurtransferase—is an emerging and promising research area. This review frames the discussion of harnessing endogenous H_2_S within the context of a non-ischemic form of cardiomyopathy, termed diabetic cardiomyopathy, and heart failure. Also, we examine the current literature around therapeutic interventions, such as intermittent fasting and exercise, that stimulate H_2_S production.

## 1. Introduction

Hydrogen sulfide (H_2_S) is an endogenously produced gaseous signaling molecule and is critical for the regulation of cardiovascular homeostasis [1,2]. It is mostly produced enzymatically in mammalian species by three enzymes in the cysteine biosynthesis pathway: cystathionine-β-synthase (CBS), cystathionine-γ-lyase (CSE), and 3-mercaptopyruvate sulfurtransferase (3-MST). New enzymes are emerging as important contributors to H_2_S production. Apart from these enzymes, H_2_S can also be produced from sulfur reservoirs (i.e., sulfane sulfur). Clinically, there is a negative association between heart failure, diabetes and H_2_S, as evidenced by the findings that lower circulating H_2_S levels are detected in plasma samples taken from diabetic and heart failure patients [3,4,5]. Therapeutic strategies aimed at increasing the levels of H_2_S are protective in models of acute myocardial ischemia-reperfusion injury and heart failure [6,7,8,9]. Numerous studies have shown the key role of H_2_S in maintaining cardiovascular homeostasis, but many of these studies rely on exogenous H_2_S donors. More studies that provide novel insight into endogenous H_2_S dynamics can help investigators understand the role of H_2_S in heart physiology.

In this review, we discuss the current understanding of H_2_S and its role in diabetic cardiomyopathy, heart failure and interventional therapies, such as intermittent fasting and exercise physiology. While significant advances have been made, the field has not fully understood how H_2_S-driven mechanisms regulate cardiac metabolism and help the heart adapt to physiological stresses (e.g., exercise). One beneficial mechanism is the H_2_S-mediated modification of cysteine residues by a process termed protein persulfidation or S-sulfhydration (PSSH), which is a reportedly prevalent modification that targets several metabolic enzymes and alters their function [10,11]. Harnessing this specific endogenous redox mechanism to rapidly modulate critical physiological and pathological signaling pathways is an exciting idea that may help to develop novel therapeutic strategies to reduce the burden of cardiovascular disease.

## 2. Overview of Hydrogen Sulfide

### 2.1. The Molecule and Post-Translational Modification 

Originally described as a toxicant, H_2_S has emerged as an essential gaseous molecule that is both clinically and physiologically relevant to the heart. While the chemistry of this molecule is reviewed elsewhere (see Filipovic et al. (2018)), in this section we will briefly revisit important properties [12]. 

H_2_S is a volatile, water-soluble molecule with the bond dissociation energy similar to thiols (90 kcal/mol vs 92 kcal/mol, respectively). This property indicates that H_2_S does not readily decompose, making it a good signaling molecule [13]. Although H_2_S is membrane diffusible with a diffusion capacity from 0.5–10 cm/s, as a reactive electrophile, intracellular factors may regulate its travel [14]. Autooxidation involving metal catalysts and oxygen that can extinguish the reactivity of the molecule and transition it into alternative oxidation states has been reported [13]. Importantly, H_2_S not only reacts with other oxidants, such as nitric oxide, but can modify oxidized thiols (e.g., sulfenic acid) to form PSSH [12,15,16]. The post-translational modification of proteins is the subject of active research to understand the molecular and physiological consequences.

For a protein to be S-sulfhydrated, the oxidation of H_2_S or the targeted thiol is required [14]. Though PSSH cannot shield proteins from oxidative damage, PSSH has a higher bond dissociation energy than other oxidative modifications, such as S-nitrosation (PSSH: 70 kcal/mol vs. 31–32 kcal/mol) [15]. This may allow PSSH to serve as a more stable signaling modification [17,18,19]. While significant advances in proteomics have identified several proteins to be dynamically S-sulfhydrated under various physiological conditions, more studies are needed to further understand the impact of PSSH and its regulation of cardiac metabolism and disease [20,21,22,23]. 

### 2.2. Mechanisms of Hydrogen Sulfide Production

An important node for H_2_S regulation is in its production. L-cysteine, generated from the cysteine biosynthesis pathway, is the fundament substrate for enzymatic H_2_S production. Three enzymes are primarily responsible for H_2_S generation: CBS, CSE, and 3-MST (Figure 1). Both CBS and CSE utilize cystathionine in a pyridoxal 5′ phosphate (PLP)-dependent reaction. On the other hand, in the cysteine catabolism pathway, L-cysteine is converted into 3-mercaptopyruvate by cysteine aminotransferase in a PLP-dependent manner and is then metabolized by 3-MST into pyruvate and H_2_S. Apart from these enzymes, others are thought to participate in transsulfuration and facilitate H_2_S production. Together, these enzymes form key nodes to produce and regulate endogenous H_2_S production. While the enzymology has been extensively studied, few studies have shown the role of these enzymes (except CSE) in cardiac biology.

CBS is a key enzyme in the reverse transsulfuration pathway that regulates the flux of sulfur from the methionine cycle to the cysteine and glutathione biosynthesis pathways. In humans, CBS is a PLP-dependent tetramer that condenses cysteine and homocysteine into cystathionine with a heme co-factor to release H_2_S (Figure 1). CBS can also condense cysteines to produce H_2_S and lanthionine. Its activity is limited by the concentration of cysteines and is regulated by S-adenosylmethionine [24,25]. Oxidative stress conditions can lead to a truncated, S-adenosylmethionine-insensitive form of CBS that has increased activity, yet other studies suggest that the oxidation of its N-terminal heme group or CXXC oxidoreductase motif may inhibit the enzyme [26,27,28]. More studies are needed to understand the role of CBS in cardiac biology and disease. While few, if any, studies have proven a cardioprotective role for CBS, mutations in this protein can lead to hyperhomocysteinemia and an increased risk for cardiovascular disease along with accelerated atherosclerosis [29]. 

CSE is the predominant H_2_S-generating enzyme in the cardiovascular system and its ability to remediate cardiac injury is under active investigation [30]. Like CBS, human CSE exists in a PLP-bound, tetrameric state that, apart from H_2_S, yields α-ketobutyrate, pyruvate, and ammonia (Figure 1) [31]. CSE is also sensitive to homocysteine levels [28]. CSE is a key player in protection from various diseases, including heart disease. Several studies have demonstrated that CSE inhibition or genetic deficiency increased infarct size and enhanced transaortic constriction (TAC)-induced heart failure severity [32,33]. Conversely, cardiac-specific CSE overexpression blunted TAC-induced heart failure progression [34]. Also, CSE and PSSH was shown to decrease in the aged heart [15]. Taken together, these studies demonstrate that CSE is an integral component of both cardiovascular disease and disease risk. 

Recently, CSE was found to be phosphorylated during atherosclerosis, leading to a decrease in H_2_S production [35]. Conversely, 17β-estradiol, the active form of estrogen, stimulates H_2_S production by CSE phosphorylation (Human Ser56) in a cyclic guanosine monophosphate/protein kinase G-dependent mechanism [36]. Also, bile salts stimulate CSE phosphorylation in endothelial cells [37]. This finding shows that H_2_S produced by CSE can be regulated by post-translational modifications (Figure 1). CSE has been shown to translocate to the mitochondria during endoplasmic reticulum stress and helps to maintain adenosine triphosphate (ATP) production. Furthermore, this translocation is Tom20 dependent and may help to regulate cystathionine levels in the mitochondria [38].

3-MST is a homodimeric, PLP-independent enzyme that receives 3-mercaptopyrvuate from cysteines metabolized by a PLP-mediated reaction via cysteine aminotransferase (CAT/GOT). Also, in the brain and kidneys, in a PLP-independent manner, D-cysteine can contribute to 3-mercaptopyruvate production via D-amino acid oxidase [39]. Upon binding, 3-MST converts 3-mercaptopyruvate into sulfane sulfur, then either thioredoxin or a thiol-based reductant liberates H_2_S and pyruvate (Figure 1). While CBS and CSE are predominately cytosolic, 3-MST is reportedly found in both the cytosol and mitochondria, suggesting that 3-MST may be a key protein that regulates mitochondrial protein PSSH. 

While phosphorylation, or other post-translational modifications have not yet been reported, or investigated, 3-MST is reportedly regulated in two ways: oxidative stress and dimerization (Figure 1). Oxidative stress from hydrogen peroxide inhibited 3-MST activity in cultured hepatoma cells, suggesting that disease which stimulate excess oxidation may suppress H_2_S production from 3-MST [40]. Furthermore, redox-sensitive cysteines within 3-MST can govern its dimerization [41,42,43]. Oxidation of 3-MST can induce dimerization rendering the enzyme inactive, however—evident from treatment with reducing agents and shielding cysteines from oxidation with an alkylating agent—3-MST monomerization maintains H_2_S production [41,43]. 

While CBS, CSE, and 3-MST are the most widely studied enzymes, others have also been identified. Rhodanese, or thiosulfurtransferase, participates in a transsulfuration reaction by catalyzing the transfer of sulfur from thiosulfate to thiol or cyanide to form persulfides or thiocyanate, respectively (Figure 1) [44]. This protein closely resembles 3-MST and may be functionally cooperative [45,46]. Interestingly, 3-MST knockout mice are reported to have a 3-fold increase in rhodanese expression that may be an adaptive response to the loss of 3-MST, however, the consequences of excessive rhodanese in these mice are not yet understood [47]. Conversely, mutation of key residues in rhodanese decreased its activity and increased 3-MST activity [48,49]. While the role of 3-MST in cardiac homeostasis and disease has been recently reported, to our knowledge, no study has yet to investigate the importance of rhodanese in cardiac biology.

Sulfane sulfurs are H_2_S reservoirs bound to proteins (e.g., PSSH) or other molecules and are important sources of H_2_S. Cysteine hydropersulfides and glutathione persulfides (GSSH) are produced endogenously and display properties for maintaining intracellular homeostasis (Figure 2) [50]. PSSH are reduced, or de-sulfhydrated, by thioredoxin to release H_2_S [23,51]. Sulfate and sulfite are also important H_2_S reservoirs. GSSH is reduced to sulfite by persulfide dioxygenase, then further oxidized to by sulfide oxidase (SO). Through sulfite production, at least with SO deficiency, GOT can also produce H_2_S by catalyzing the deamination of cysteine sulfinic acid, the first product of oxidative cysteine metabolism [52]. While there have been advances in H_2_S biology, more studies are needed to fully comprehend the complexity of this redox molecule, especially regarding its role in cardiac metabolism. 

## 3. Hydrogen Sulfide in Heart Metabolism and Cardiac Disease

### 3.1. Overview

The heart relies on fatty acids to generate energy, while other organs utilize glucose as their primary energy substrate [53,54]. The energy demand of the heart requires the use of fatty acids, because this substrate yield more ATP molecules than glycolysis [55]. Long-chain fatty acids are transported into cells by fatty acid translocase, CD36 [56]. Once inside the cell, fatty acid oxidation (FAO) is initiated with the transport of long-chain fatty acids that are modified by long-chain acyl-co-enzyme A (CoA) synthetase (LACS) and carnitine palmotyltransferase 1 (CPT1). LACS cleaves an acyl-CoA molecule and delivers it to CPT1 [57]. 

CPT1 is a rate-limiting enzyme of FAO that controls the entry of acyl-CoA into the mitochondria. In cardiomyocytes, CPT1b is the predominant isoform, accounting for 98% of CPT1 activity, while in other organs CPT1a (e.g., liver) and CPT1c (neurons) are more active [58,59,60,61,62]. CPT1 converts acyl-CoA into acylcarnitine and transports the product across the outer mitochondrial membrane. Then, carnitine acylcarnitine translocase (CACT) allows acylcarnitine to traverse the inner mitochondrial membrane and react with CPT2. With CPT2, acylcarnitine is converted back into acyl-CoA and enters the tricarboxylic acid (TCA) cycle [57].

Briefly, the TCA cycle is amongst the most well-studied and most important metabolic cycles in biology. The churn begins with the condensation of acyl-CoA into citrate, then the cycle carries citrate to oxaloacetate. Along the way, the TCA cycle produces two essential reducing equivalents—nicotinamide adenosine dinucleotide + hydrogen (NADH) and flavin adenosine dinucleotide + 2 hydrogen (FADH_2_)—along with other metabolites that participate in other metabolic systems. Both NADH and FADH_2_ feed electrons into the Electron Transport Chain that ultimately reduces oxygen and drives the migration of hydrogen over the mitochondrial membrane, power ATP synthase, which generates ATP (Figure 2). From one C_18_ fatty acid, FAO can yield 120 ATP molecules versus the same amount of ATP from three glucose molecules [55].

The heart beats ceaselessly throughout a human’s, or animal’s, life and generates pressures needed to overcome vascular resistance. This process requires enough energy to sustain homeostasis for both the cardiomyocytes and the other cells in the body. FAO gives the heart the energy it needs to function, however disease can the shift the metabolic substrate used and often forces the heart to utilize glycolysis instead of FAO [63]. H_2_S may be a way to preserve cardiac metabolism and remediate cardiovascular disease. 

### 3.2. Hydrogen Sulfide Regulation Integrated with Mitochondrial Respiration

Redox homeostasis requires both the production of reactive molecules that can alter cellular function and a means of controlling that production. H_2_S regulation is intimately linked with metabolism and mitochondrial respiration (as reviewed by Paul et al. (2021)) [64]. Although H_2_S can suppress metabolism at high doses, recent studies show that low dose H_2_S is a metabolic stimulator, in part by providing electrons for mitochondrial respiration [65,66,67,68]. At low concentrations, H_2_S can serve as an inorganic electron donor—as it does in microorganisms—to Coenzyme Q with the help of sulfide quinone oxidoreductase (SQR) [64,69]. 

Along with persulfide dioxygenase and rhodanese, SQR is a key component of the Sulfide Oxidation Unit, which is responsible for the catabolism of free H_2_S (Figure 2) [69,70]. H_2_S is oxidized by SQR to yield two electrons that are transferred to FAD, then Coenzyme Q, which reduces Complex III [71]. This reaction also produces thiosulfate or sulfate that reacts with reduced glutathione to form GSSH. This new persulfide on glutathione is then oxidized to sulfide by persulfide dioxygenase. Sulfide can be further metabolized by either sulfide oxidase to sulfate or rhodanese to a thiosulfate. Thiosulfate can be catabolized back into H_2_S by 3-MST or rhodanese [72,73]. The cyclical nature of these reactions not only controls the toxic accumulation of H_2_S but may help to maintain a physiologic pool of sulfur that can be used for important redox reactions and other essential molecular processes. 

Various cardiac diseases are linked to decreased H_2_S production and availability. Under active investigation are mechanisms that regulate H_2_S production and the ways they are impaired during disease. Below, we focus on two cardiac diseases well understood to suppress H_2_S production and availability, and their linked to H_2_S-mediated metabolic regulation: diabetic cardiomyopathy and heart failure. 

### 3.3. Diabetic Cardiomyopathy and Hydrogen Sulfide

The prevalence of obesity, insulin resistance, diabetes, and dyslipidemia are increasing worldwide. People who have these disorders, collectively called metabolic syndrome, are twice as likely to develop heart failure and have worse prognoses after cardiovascular disease development [74]. In the U.S., obesity has become a major risk factor for metabolic syndrome development. From 2013–2019, 34% of overall children (2–19 years old) and 38% of adults in the U.S. were obese. Notably, the highest prevalence was amongst Hispanics [75]. Studies implicate “Western” diets, or high-fat diets (HFD), as a key driver of obesity [76]. Most patients with metabolic syndrome have hypertriglyceridemia and increased plasma levels of fatty acids. These lipids are taken into the heart, where due to an overload in storage capacity and utilization, accumulate and become lipotoxic to the cardiomyocytes. This induces a non-ischemic form of cardiomyopathy termed lipotoxic or diabetic cardiomyopathy (DCM) that can progress to heart failure [74,77]. 

Metabolic syndrome is strongly associated with a significant decrease in H_2_S availability and protein PSSH [3,4]. This syndrome leads to DCM due to the accumulation of lipids that become toxic. Oxidative stress and inflammation merge with toxic lipid accumulation to form more oxidants that overwhelm the metabolic system of the heart and kill the irreplaceable cardiomyocytes [74]. Cell death then leads to pathologic hypertrophy and can decompensate into heart failure [78,79]. This non-ischemic form of cardiomyopathy is linked to dysregulated metabolism and suppressed H_2_S production.

Clinically, there is a negative association between diabetes and H_2_S, evident from the findings that lower circulating H_2_S levels are detected in plasma samples taken from diabetic patients [3,4]. Preclinical models, specifically HFD-fed mice or fatty acid-treated cell models, have replicated these findings. Recent animal work showed that lost H_2_S bioavailability (e.g., lowered CSE expression, reduced protein PSSH, and impaired H_2_S production) contributes to the etiology of metabolic syndrome [80,81,82,83,84,85,86]. Additionally, cell models treated with toxic lipids, such as palmitate, reported impaired H_2_S production and reduced protein PSSH [87]. Many studies have shown that exogenous H_2_S can protect against excessive heart injury, including DCM [88,89,90]. While H_2_S donors are a promising form of therapy, stimulating endogenous H_2_S production is an innovative strategy that may improve target specificity. 

Few studies have studied the underlying mechanisms that govern H_2_S-mediated regulation of metabolism; however, this is an active area of research. H_2_S can not only donate to the mitochondrial respiration, but it can also modify ATP production. Under stress conditions, mitochondrial CSE translocation was shown to increase mitochondrial H_2_S and enhanced ATP production in smooth muscle cells [38]. ATP5A1, a subunit of ATP synthase, is also known to be S-sulfhydrated in a CSE-dependent manner [91]. In streptozotocin-induced, diabetic rats, 3-mercaptopyruvate—the substrate for 3-MST—was injected intraperitonially and elevated circulating H_2_S, along with stimulating oxygen consumption from liver mitochondria [92]. These studies suggest that elevating mitochondrial H_2_S may be a protective adaptation to regulate bioenergetics. Apart from mitochondrial respiration, our group has shown that H_2_S can stimulate carbohydrate and lipid metabolism. Through the activation of adenosine monophosphate kinase (AMPK) by S-sulfhydrating and inhibiting scaffolding protein phosphatase 2A (PP2A), the heart upregulated metabolic genes (Figure 1) [93]. Recently, Bithi et al. (2019) described a cardiac and endothelial S-sulfhydryl proteome [20,94]. Several proteins involved in metabolism were identified, yet the impact of PSSH on enzymatic activity and cardiac metabolism remains to be understood.

A way that H_2_S can regulate metabolism is by modulating transcription factors activation, specifically those responsible for the upregulation of metabolic genes (Figure 3). Peroxisome proliferator-activated receptors (PPAR) are major transcription factors that control the expression of FAO and uptake genes in all organs [95,96]. Both PPAR-δ and PPAR-γ are modified and activated by H_2_S, inducing lipid metabolism [97,98]. Our group has shown that peroxisome proliferator-activated receptor gamma coactivator (PGC)-1α, a key co-regulator of PPAR signaling is induced by H_2_S treatment through the stimulation of AMPK [93]. Hypoxia-inducible factor (HIF)-1α is an essential transcription factor that stimulates the expression of glycolytic genes in response to hypoxia or oxidation [99]. Under normoxic conditions, CBS-dependent H_2_S inhibits prolyl hydroxylase 2, which helps target HIF-1α for degradation, thus allowing HIF-1α to accumulate and induce gene expression [100,101]. These important proteins regulate the expression of metabolic genes demonstrating that H_2_S can even regulate metabolism at the transcriptional level.

Few studies have focused on the impact that dysregulated H_2_S can have on cardiac metabolism and disease. The field needs more innovative experiments and approaches that capture the dynamics of H_2_S production and signaling to understand the beneficial nature of this gaseous molecule. Harnessing this endogenous donor pool may help reduce the burden of cardiovascular disease by adding specificity to the existing body of exogenous H_2_S donors. A way to study these dynamics may be to study the underlying molecular mechanism in the development of metabolic syndrome and the interventions that reduce cardiovascular disease risk from metabolic disorders.

### 3.4. Heart Failure

According to the annual American Heart Association statistics, about 6.2 million Americans over the age of 20 were affected with heart failure between 2013–2016, and that number is expected to rise with an aging population. Heart failure is projected to increase by 46% between 2012–2030 [102]. Racial, ethnic, and gender disparities exist with heart failure. In the Health, Aging and Body Composition Study, the risk of developing heart failure the elderly black population—attributed to modifiable risk factors, such as smoking and blood pressure—was about 68% compared to 49% for the elderly white population [103]. The Hispanic population, which also has a high cardiometabolic risk—as previously described—also has a high risk of developing heart failure [104,105]. While women tend to have more co-morbidities, such as diabetes and high blood pressure, heart failure mortality prognosis is better for women compared to men [106,107]. Reducing the burden of heart failure in the aging population is imperative and molecular insights may help to develop preventative interventions that can lower heart failure risk.

Heart failure—a consequence of DCM—is characterized as a significant loss of cardiac output due to a substantial alteration in heart structure [108]. The etiology of heart failure can come from various sources, mostly from events that injure the heart and results in the death of irreplaceable cardiomyocytes. As previously described, DCM is a non-ischemic form of cardiomyopathy and over time results in heart failure, however ischemic injury also reduces cardiac function. Myocardial infarctions—cardiac muscle death usually resulting from an occlusion of the coronary arteries which feed the heart—and subsequent reperfusion is an example of ischemic injury that can lead to heart failure [109]. The cardiomyocytes of the injured heart swell and undergoes pathological hypertrophy to compensate for a loss in cardiac output, however this adaptation is unsustainable [110,111]. Over time, the muscle degrades, the heart scars, and decompensates into heart failure [78,79].

Heart failure manifests as symptoms of dyspnea (labored breathing) and pulmonary congestion (i.e., edema) among others, resulting from significant physiological alterations [108]. A consequence of reduced cardiac output is a drop in blood pressure. This stimulates the brain to increase β-adrenergic signaling by releasing more norepinephrine and epinephrine from the adrenal glands. Meanwhile, the kidneys initiate the renin-angiotensin system to induce vasoconstriction, which can lead to kidney disease [112,113]. Both adaptations increase blood pressure to healthy levels, attempting to elevate organ perfusion and compensating for the loss of cardiac output. However, the heart is not as it once was; this “high” blood pressure further strains the weakened heart, killing more cardiomyocytes [110,114]. 

The role of endogenous H_2_S production and regulation is not completely understood in heart failure. Heart failure can be categorized into heart failure with reduced or preserved ejection fraction (HFrPF or HFpPF, respectively) [115,116]. While reports are emerging that focus on HFpPF, a large body of literature has described the molecular mechanisms involved in HFrPF. In severe, end-stage heart failure patients, there is a marked reduction in circulation H_2_S that is reproduced in pressure-induced heart failure animal models, such as TAC [5]. Furthermore, several studies have shown that pharmacological donors of H_2_S, such as sodium sulfide, SG-1003, among others, preserve cardiac function in pressure-induced and ischemia-induced heart failure animal models [9,117,118,119]. Heart failure in CSE deficient mice was more severe compared to WT controls [34]. These studies strongly implicate H_2_S as a key mechanism of heart failure.

In animal models, heart failure can be induced by pressure overload with TAC—banding of the aorta to simulate excess arterial pressure. Also, surgical occlusion of a coronary artery—cardiac vessels that feed the heart—can cause ischemia. Our group showed that sodium sulfide protected hearts from ischemic injury in a Nuclear factor E2-related factor 2 (Nrf2)-dependent manner (Figure 3). H_2_S-mediated activation of this transcription factor, which is responsible for upregulating antioxidant responses, sustained proteosome activity, thus eliminating dysfunctional proteins and preserving cellular homeostasis [119,120]. Moreover, H_2_S stimulated pro-angiogenic factors, such as nitric oxide bioavailability and vascular endothelial growth factors, increasing cardiac vascular density [117]. In culture cardiomyocytes, H_2_S attenuated hypertrophy and stimulated glycolysis by upregulating glucose transporter 4, along with increasing pyruvate kinase and succinate dehydrogenase activity [121]. Also, age-dependent cardioprotection against ischemia-reperfusion injury was reported in young 3-MST knock out mice, while 18-month-old mice showed hypertension and cardiac hypertrophy [122]. Taken together, H_2_S donors preserve cardiac structure and function, however endogenous H_2_S—evident from CSE deficient mice—also contributes to protection.

While H_2_S donors remain a vital strategy to efficently deliver this potent and protective molecules, we propose that exploiting the endogenous, cardiac H_2_S system can also be an integral and specific way of remediating heart failure and other cardiovascular diseases. Historically, the benefits of H_2_S of heart disease were demonstrated by donors, because—until recently—few stimulators of endogenous H_2_S were known. These H_2_S donors are non-specific because they release H_2_S throughout the body or an organ. H_2_S-producing enzymes are naturally positioned throughout and likely target specific, redox-sensitive proteins initiate a cardiac stress response, thus confer protection against disease. Targeting these H_2_S enzymes and modifying their activity—without genetic manipulation, such as removal or overexpression—may be a way to develop targeted therapeutics. However, few studies have investigated endogenous H_2_S regulation during disease. Of the few reports, cultured hepatocytes supplemented with 3-mercaptopyruvate—the substrate for 3-MST—stimulated 3-MST- and SQR-dependent H_2_S production. This treatment increased mitochondrial bioenergetics, yet the experiments were not set in the context of disease [123]. The role of endogenous H_2_S in the heart remains to be fully understood, yet there is clearly a role for H_2_S enzymes, such as 3-MST, in maintaining cardiac homeostasis during disease. Physiological stimulators of cardiac H_2_S are emerging, and these stimulators are well-known interventional strategies, which reduce the risk of cardiovascular disease.

## 4. Physiological Stimulators of Hydrogen Sulfide Production

### 4.1. Overview of Cardiovascular Disease Risk

Evaluating cardiovascular disease risk is an important metric that can help predict the likelihood of developing cardiovascular disease and is influence by a number of modifiable risk factors. Body weight, insulin sensitivity (i.e., diabetes), low-density lipoprotein (LDL) cholesterol, and physical inactivity, along with cigarette smoking, are well-known risk factors that can affect cardiovascular disease risk [124]. Because they are considered modifiable risk factors, interventions can significantly help to prevent, or exacerbate, the development of cardiovascular disease and reduce the public health burden of these diseases. In this section, we review two interventions that are linked to stimulating H_2_S production. The molecular mechanisms that confer the cardiovascular benefits of intermittent fasting and exercise are emerging and H_2_S is intimately associated with these intervention strategies.

### 4.2. Intermittent Fasting and Hydrogen Sulfide

The ability of intermittent fasting (IF) to reduce cardiovascular disease risk, especially from metabolic diseases is well-described and extends beyond reducing body weight [125,126,127,128,129]. There are three different methods of IF: (1) alternate-day fasting (ADF), where feeding is restricted for one day, then *ad libitum* feeding another day; (2) Modified ADF (MADF), such as with “IF 5:2”, feeding is *ad libitum* for 5 days, then restricted for 2 days [130]. Finally, (3) time-restricted feeding (TRF) is where feeding is *ad libitum* for a few hours, then fasting occurs for another few hours [131,132]. TRF is commonly done with overnight fasting models. These methods are often coupled with caloric restriction, which limits the energy obtained from *ad libitum* by a certain percentage [131]. In human and animal models studies have aimed at assessing the benefits of IF on cardiovascular disease prevention and remediation.

IF strategies can reduce cardiovascular disease risk and several studies have demonstrated this effect (as reviewed by Crupi et al. (2020)) [133,134]. Apart from weight loss, caloric restriction reduced cardiovascular disease risk markers, such as LDL cholesterol in nonobese subjects [126]. IF can also help reduce cardiovascular disease risk in at-risk populations, such as pre-diabetic, overweight men. In a randomized study, patients were grouped into a TRE or control group. The investigators concluded that TRE, particularly early morning restriction, improved insulin sensitivity, blood pressure, and oxidative stress, despite no change in body weight [135]. While diet adherence is a challenge, a study found that self-administered ADF reduced blood pressure, heart rate, arterial pulse pressure, and body fat over a long period of time [136]. More research is needed to understand the mechanisms that contribute to the benefits of IF. 

An important feature of intermittent fasting is metabolic switching. In other organs, such as the liver, reduced caloric intake activates lipolysis from adipose tissues, allowing for FAO [137,138]. This metabolic switch allows for greater metabolic flexibility and more energy production [139]. From FAO, acyl-CoA feeds the TCA cycle, but when acyl-CoA concentrations exceed citrate synthase activity, ketogenesis occurs [140]. These ketone bodies, apart from participating in metabolic signaling, are used as energy substrates [141]. Ketone bodies, specifically β-hydroxybutyrate, activate cellular stress defenses—for instance, antioxidant and anti-inflammation—improving disease resistance [142,143]. The effect of IF on cardiac metabolism remains to be investigated, however insight from other organs can help us understand how the heart may respond. 

The significance of ketogenesis on cardiac health, especially during disease is emerging in the field, due, in part, to its role in IF. While ketone bodies cannot provide as much energy as FAO, they can yield more ATP for less oxygen than glycolysis and can help prevent mitochondrial uncoupling—referred to as the “thrifty fuel” hypothesis” [54]. β-hydroxybutyrate—elevated by IF and heart failure—is a potent repressor of oxidative stress, and infusion was shown to enhance cardiac output in animals and HFrEF patients [142,144,145]. Recently, HFpEF patients were reported to have less β-hydroxybutyrate—a ketone body that accounts for about 75% of circulating ketones—however in a HFpEF animal model, β-hydroxybutyrate treatment reduced fibrosis, lung edema and other HFpEF markers [146]. While in human U937 monocytes, β-hydroxybutyrate had no effect on CSE expression or activity, hyperketonemia from acetoacetate decreased H_2_S production [147]. Since IF increases β-hydroxybutyrate and given the protective ability of this ketone, IF may be a beneficial practice to help reduce cardiovascular disease risk. While a direct link between ketone metabolism and H_2_S in the heart has not yet been made, more studies are needed to show this association. 

IF is an important stimulator of H_2_S production in various organs, but few studies have examined cardiac H_2_S production. From redox biology, we can deduce that H_2_S may be a way to fine tune cardiac metabolism. Gao et al. (2020) described that under oxidative stress conditions, excess H_2_S can stimulate a thiol switch from S-glutathioinylation to PSSH. According to the article, this switch creates a new redox active thiol [22]. Further studies are needed to determine if this event can be a product of metabolic status, but this may be possible. Dietary caloric restriction enhanced H_2_S production and protein PSSH in various organs in a CSE-dependent mechanism [20,148]. Also, a hyperhomocysteinemia model—CSE and CBS deletion—abrogated fasting-induced protection from myocardial ischemic injury [149]. These studies suggest that H_2_S may be a key mediator of the benefits afforded from IF. 

### 4.3. Exercise and Hydrogen Sulfide

In addition to IF, exercise also modulates cardiovascular diseases risk factors [150,151,152]. Physical inactivity is a risk factor for cardiovascular disease and exercise is a potent preventative factor that can reduce the risk of heart disease [153,154]. A recent study showed that physically inactive behaviors, established during childhood, can continue through adulthood and impact the cardiovascular disease risk. The study also found that even a little bit of physical activity can lower disease risk [155]. Establishing an exercise routine even with older adults can be beneficial. In 2004, Brach et al. found that in men and women age 70–79, acute (20–30 min), moderate exercise improves physical functions [156]. Exercise can be categorized by intensity. Moderate- to high-intensity, aerobic training is known to benefit cardiac health, especially for those with the risk factors for cardiovascular disease [157,158,159]. Coupled with IF, or caloric restriction, exercise can greatly improve heart health. Many of the underlying mechanisms, specifically relating to H_2_S production, are still under active investigation [158]. 

As opposed to pathological hypertrophy in heart failure, exercise induces physiological hypertrophy. This is a beneficial adaptation that increases cardiac output and allows for greater muscle perfusion, raising oxygen delivery [160]. Exercise stimulates skeletal muscle metabolism, specially by raise glycolytic flux from glycogen stores depletion and higher insulin sensitivity. These metabolic changes also alter muscular pH and helps to liberate oxygen bound to hemoglobin. Exercise also raises respiratory ventilation that can match elevated blood perfusion, which allows the removal of excess carbon dioxide—generated from increased muscle metabolism—from the blood and the infusion of more oxygen. These adaptations, while temporary, can have lasting benefits on cardiovascular function. 

Preclinical, animal models can provide insight into the beneficial molecular mechanisms of exercise. Apart from intensity, exercise can also be categorized by modality, frequency, and duration [161]. An example of an exercise modality includes dynamic exercise, which employs various muscle groups, along with static and resistive training [162]. Investigators often us treadmill running, voluntary wheel running, or swimming to study the benefits of dynamic exercise, because these experiments require multiple muscle through throughout the animal’s body [163,164]. A model to assess the benefits of resistive training. Briefly, for this method, the animal is suspended and positioned on its hind legs, then electrical stimulation coerces the muscle to flex and the animal squats against a weight [165,166]. All have their advantages and disadvantages (as reviewed by Feng et al. (2019)), but they have allowed researchers to study the molecular benefits of exercise.

Few studies have explored the direct mechanisms that regulate H_2_S production during exercise, but these few studies suggested that exercise stimulates H_2_S. These studies also demonstrated that H_2_S dynamics are an essential component of exercise physiology. Several studies have shown that sedentary patients with varying intensity and frequency of training modified homocysteine concentrations—a substate for CBS, which can initiate the reverse transsulfuration pathway and trigger H_2_S production [167,168,169,170,171]. Animal models have shown that exercise training elevates the expression of H_2_S-generating enzymes, such as CSE and 3-MST. Moderate-intensity exercise in aged rats increase CSE and 3-MST expression in the heart, thus increasing H_2_S availability, while lowering oxidative stress and fibrosis [172]. In ovariectomized rats, which have reduced CSE expression, exercise augmented the damage induced by heart failure and restored CSE expression [173]. Although H_2_S production was not evaluated, these results suggest a role of H_2_S by altering the flux of metabolites in the reverse transsulfuration pathway. 

Coupling exercise with dietary status is often used by researchers to study H_2_S production, which is likely a more clinically relevant design to examine the molecular pathways involved with preventing cardiovascular disease [174,175,176]. In mice fed an HFD for 20 weeks, exercise attenuated metabolic syndrome development and blunted cardiac dysfunction. In the exercise/HFD-fed group, left ventricular H_2_S increased, along with CBS and CSE upregulation [152]. Additionally, endothelial CSE influences exercise capacity. Genetic CSE overexpression and deletion, respectively, improves and impairs running distance with a treadmill exercise test [177]. Insight can also be gleamed from non-cardiac tissue. Exercise training of HFD-fed mice enhanced CBS, CSE, and 3-MST expression and H_2_S bioavailability in liver tissue, while elevating antioxidant capacity [178]. In a model of chronic kidney disease, where sections of the kidney were removed from rats, exercise training for 8-weeks not only reduced oxidative stress, but also increased H_2_S levels [179]. These studies show that exercise is an important intervention strategy to protect the heart from cardiovascular disease linked to metabolic syndrome, in part by stimulating cardiac H_2_S production and that this mechanism is necessary to sustain exercise capacity. 

## 5. Conclusions

Interventions and therapeutics to reduce the burden of cardiovascular disease are essential for improving public health. While H_2_S donors are potent treatments for cardiac injury, stimulating endogenous H_2_S production may be a novel and natural strategy to prevent cardiovascular disease. Several studies have shown that heart disease negatively impact H_2_S dynamics in the heart, and this impairment is linked to bioenergetics. Preventative strategies proven to reduce the risk of cardiovascular disease developing—such as intermittent fasting and exercise—are shown to stimulate and regulate H_2_S production. In turn, H_2_S can regulate bioenergetics that beneficially promote cardiac health. Further study of these dynamics in the heart may unlock a new perspective on cardiac metabolism and may help to remediate cardiovascular disease.

## Figures and Tables

**Figure 1 antioxidants-10-00383-f001:**
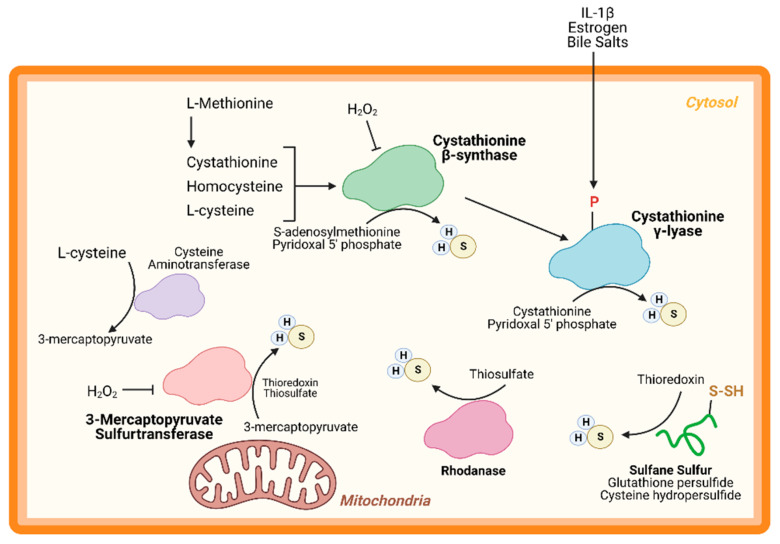
**Hydrogen sulfide (H_2_S) Production.** H_2_S is predominately produced by three enzymes—cystathionine-β-synthase, cystathionine-γ-lyase, and 3-mercaptopyruvate sulfurtransferase—and can be generated from other processes, such as sulfide quinone oxidoreductase and sulfane sulfur reservoirs (e.g., glutathione/cysteine (hydro)persulfides). H_2_S can modify free cysteines on enzymes, a reaction known as S-sulfhydration, to alter metabolic pathways, such as glycolysis, and induce up-regulation of metabolic genes via transcription factor activation.

**Figure 2 antioxidants-10-00383-f002:**
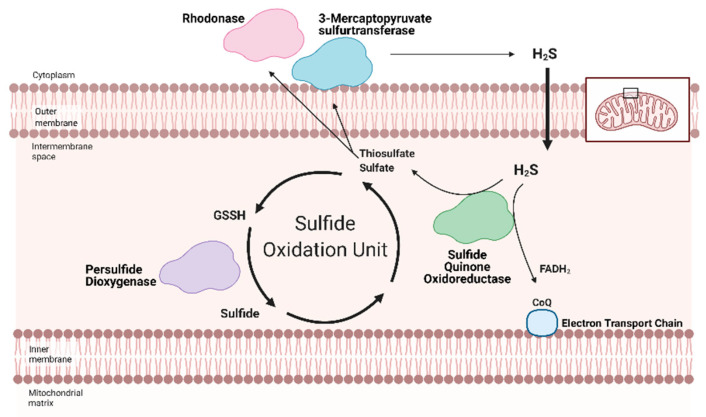
**Hydrogen Sulfide (H2S) catabolized by the Sulfide Oxidation Unit.** Free H2S can be removed from the cell either by storage in sulfane sulfur pools or the Sulfide Oxidation Unit. H2S catabolism is linked to mitochondrial respiration via sulfide quinone oxidoreductase (SQR). Nicotinamide adenosine dinucleotide + hydrogen (NADH) and flavin adenosine dinucleotide + 2 hydrogen (FADH_2_) reduce Complex I and II, respectively. The electrons travel across the Electron Transport Chain, through Complex III and cytochrome c (Cyt c) to ultimately reduce oxygen in Complex IV. This flow generates a hydrogen gradient that drives adenosine triphosphate (ATP) synthase and allows for ATP generation from adenosine dinucleotide (ADP). SQR, a key part of the Sulfide Oxidation Unit, extracts electrons from H2S and reduces co-enzyme Q (CoQ), which transfers the electrons to Complex III, contributing to ATP production. As a byproduct, SQR also produces thiosulfate and sulfate—the former can be converted back into H2S by 3-mercaptopyruvate sulfurtransferase. As part of the Sulfide Oxidation Unit, thiosulfate and sulfate react with glutathione to form glutathione persulfide (GSSH), which is then converted into sulfide and recycled back into thiosulfate. Adapted from “Electron Transport Chain”, by BioRender.com (accessed on 24 February 2021). Retrieved from https://app.biorender.com/biorender-templates(accessed on 24 February 2021).

**Figure 3 antioxidants-10-00383-f003:**
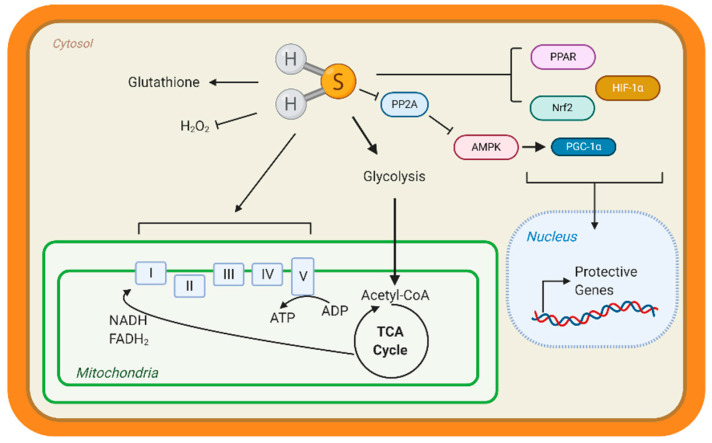
**Hydrogen sulfide (H2S) Metabolic Targets in the Heart.** Few studies have identified metabolic targets of H2S, but those that have been identified are key for cardiac metabolism. H2S can modify free cysteines on enzymes, a reaction known as S-sulfhydration, to alter metabolic pathways, such as glycolysis, and induce up-regulation of metabolic genes via transcription factor activation. Peroxisome proliferator-activated receptors (PPAR), hypoxia-inducible factor 1α (HIF-1α), and nuclear factor E2-related factor 2 (Nrf2) are transcription factors known to be S-sulfhydrated and regulate heart metabolism. H2S also modified the complexes of the Electron Transport Chain and can stimulate mitochondrial respiration. H2S is considered an antioxidant molecule because it can react with glutathione and hydrogen peroxide (H2O2). Finally, our group found that S-sulfhydration represses protein phosphatase 2A (PP2A) activity, allowing for the activation of adenosine monophosphate kinase (AMPK) and phosphorylation of the transcription factor peroxisome proliferator-activated receptor γ coactivator-1α (PGC-1α) to induce gene expression.

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
