# Peer review of "Harnessing the Benefits of Endogenous Hydrogen Sulfide to Reduce Cardiovascular Disease"

_antioxidants, 2021, doi:10.3390/antiox10030383_

Round 1
Reviewer 1 Report
This is a fine overview of the role of H2S in cardiovascular system. I have some corrections and suggestion that I think authors should take into consideration.
Page 1. SSH is really not the best abbreviation as it is also a chemical formula of disulfide radical. I would recommend to the authors to use PSSH for protein S-sulfhydration. In addition although widely used, S-suflhydration is maybe not the best term. I would leave it to the authors to decide what they want to do, but I think that they should mention that it is also called protein persulfidation.
Page 2.
Line 47. H2S is not a radical. I would remove the first two sentences.
Line 58. H2S does not react with free thiols (see Filipovic et al. 2018). It reacts with oxidized thiols, mainly sulfenic acids (see Zivanovic et al, 2019). The mentioned reaction with NO requires citation (Eberhardt et al, Nature Comm, 2014).
Line 62, this sentence has non chemical meaning. Please rephrase or remove.
Line 64. I would stop this sentence at “required”. The second part of the sentence is not clear. Oxidation state of sulfur in H2S and HS- is the same.
Line 66-68. PSSH does protect from oxidative damage as shown in great lengths by Zivanovic et al, Cell Metabolism, 2019.
Line 74-75. While CSE and CBS do use cysteine generated in cysteine biosynthesis pathway, MST is a part of cysteine catabolism pathway so for the sake of accuracy I would suggest to modify this sentence.
Page 3: CSE has been also shown to decrease in aging heart (PSSH as well) Zivanovic et al, Cell Metabolism, 2019.
Line 154. PSSH (and therefore sulfane sulfur source) can also serve as a source of H2S. PSSH get de-persulfidated by thioredoxin to release H2S (Doka et al, Sci Adv, 2016, Wedmann et al, Chem Sci, 2016).
Page 6
Line 218. Please include reference Vitvitsky et al, ACS Chem Biol, 2018 where this is actually demonstrated. Neither of those two references 63,64 show that H2S stimulates respiration. Furthermore, this reference together with Arndt et al. JBC, 2017 show that in hypoxia there is accumulation of H2S due to inability of SQR to remove H2S.
Author Response
This is a fine overview of the role of H2S in cardiovascular system. I have some corrections and suggestion that I think authors should take into consideration.
Page 1. SSH is really not the best abbreviation as it is also a chemical formula of disulfide radical. I would recommend to the authors to use PSSH for protein S-sulfhydration. In addition although widely used, S-suflhydration is maybe not the best term. I would leave it to the authors to decide what they want to do, but I think that they should mention that it is also called protein persulfidation.
Author Response:We thank the review for this correction and suggestion. We have changed the abbreviation through the manuscript and have included the term protein persulfidation (Page 1; line 40)
Page 2.
Line 47. H2S is not a radical. I would remove the first two sentences.
Author Response:We thank you for the clarification and have removed both sentences.
Line 58. H2S does not react with free thiols (see Filipovic et al. 2018). It reacts with oxidized thiols, mainly sulfenic acids (see Zivanovic et al, 2019). The mentioned reaction with NO requires citation (Eberhardt et al, Nature Comm, 2014).
Author Response:We thank you for the references and have added them to the manuscript (Page 2; line 60).
Line 62, this sentence has non chemical meaning. Please rephrase or remove.
Author Response:We thank you for the clarification and have removed this sentence.
Line 64. I would stop this sentence at “required”. The second part of the sentence is not clear. Oxidation state of sulfur in H2S and HS- is the same.
Author Response:We thank you for the clarification and have stopped this sentence at “required” and removed the latter half of the sentence (Page 2; Lines 62-63).
Line 66-68. PSSH does protect from oxidative damage as shown in great lengths by Zivanovic et al, Cell Metabolism, 2019.
Author Response:We thank you for the reference and have added it to the manuscript (Page 2; lines 63-65).
Line 74-75. While CSE and CBS do use cysteine generated in cysteine biosynthesis pathway, MST is a part of cysteine catabolism pathway so for the sake of accuracy I would suggest to modify this sentence.
Author Response:We thank you for the clarification and have included the reference to the “cysteine catabolism pathway” to the manuscript (Page 2; line 75).
Page 3: CSE has been also shown to decrease in aging heart (PSSH as well) Zivanovic et al, Cell Metabolism, 2019.
Author Response:We thank you for this reference and have included it in the manuscript and along with a concluding sentence to the paragraph (Page 4; lines 111-113).
Line 154. PSSH (and therefore sulfane sulfur source) can also serve as a source of H2S. PSSH get de-persulfidated by thioredoxin to release H2S (Doka et al, Sci Adv, 2016, Wedmann et al, Chem Sci, 2016).
Author Response:We thank you for these references and have added them to the manuscript (Page 5; Line 155-156).
Page 6
Line 218. Please include reference Vitvitsky et al, ACS Chem Biol, 2018 where this is actually demonstrated. Neither of those two references 63,64 show that H2S stimulates respiration. Furthermore, this reference together with Arndt et al. JBC, 2017 show that in hypoxia there is accumulation of H2S due to inability of SQR to remove H2S.
Author Response:We thank you for these references and have included them in the manuscript (Page 7; Line 220)
Reviewer 2 Report
Main comment
The beneficial effects of H2S on cardiovascular homeostasis is gaining strength as an established paradigm, hence studies and reviews on the subject like this are welcome.
The main justification of this review is that much is known and is being studied on the use of increased levels of H2S for the treatment of cardiopathies with H2S-releasing drugs, but little has been done on strategies based on the stimulation of endogenous H2S production. The main limitation is that the complex interplay between the three main enzymes involved in the metabolic production of H2S makes it difficult to design a therapeutic strategy based on overexpression or removal of enzymes.
The reasoning would then be that if one can increase H2S levels with specific H2S releasing drugs it does not pay to try and manipulate complex metabolic pathways that could present collateral undesirable effects. This reasoning would make this review useless.
However, the authors contention is that already proven preventive strategies to reduce the risk of cardiovascular disease (intermittent fasting and exercise) do actually act through elevating H2S levels and regulating bioenergetics. The authors emphasize the involvement of H2S dynamics in the beneficial effects of dietary restriction, intermittent fasting and exercise in cardiac health. Despite the scarcity of evidences, which they acknowledge, this contention, well supported by the bibliographic references, may sound convincing. However, the authors do not overtly claim to dismiss the use of H2S elevating drugs resulting in a rather plain text. A clear statement to define the authors’ positioning would strengthen the impact of the review.
Minor comments
- Lines 64-65: not easy to understand the meaning of this sentence.
- Figure 1: This illustration employs conventional typology used to write chemical/enzymatic reactions but the intended meaning is unconventional. Due to this incoherence, the figure is not self-explained. A better and more elaborated scheme or illustration would help.
- Lines 206-207: this is an incorrect calculation: the canonical production of ATP from C18 stearic acid through beta oxidation and respiratory metabolism is 120 ATP/mol.
- Figure 2: This scheme attracts the attention excessively toward the electron transport chain and the whole process of mitochondrial respiration, which is shown in detail, and little is dedicated to the main point in this review: the fate of H2S. The lower part should be abbreviated or schematized and the “sulfite oxidation unit” should be expanded.
- Typographical errors have to be corrected: for instance, lines: 167, 170, 251, 321, 344, 404, 490, 502, 504, 505, 645.
Author Response
The beneficial effects of H2S on cardiovascular homeostasis is gaining strength as an established paradigm, hence studies and reviews on the subject like this are welcome.
The main justification of this review is that much is known and is being studied on the use of increased levels of H2S for the treatment of cardiopathies with H2S-releasing drugs, but little has been done on strategies based on the stimulation of endogenous H2S production. The main limitation is that the complex interplay between the three main enzymes involved in the metabolic production of H2S makes it difficult to design a therapeutic strategy based on overexpression or removal of enzymes.
The reasoning would then be that if one can increase H2S levels with specific H2S releasing drugs it does not pay to try and manipulate complex metabolic pathways that could present collateral undesirable effects. This reasoning would make this review useless.
However, the authors contention is that already proven preventive strategies to reduce the risk of cardiovascular disease (intermittent fasting and exercise) do actually act through elevating H2S levels and regulating bioenergetics. The authors emphasize the involvement of H2S dynamics in the beneficial effects of dietary restriction, intermittent fasting and exercise in cardiac health. Despite the scarcity of evidences, which they acknowledge, this contention, well supported by the bibliographic references, may sound convincing. However, the authors do not overtly claim to dismiss the use of H2S elevating drugs resulting in a rather plain text. A clear statement to define the authors’ positioning would strengthen the impact of the review.
Author Response: We thank you for this comment and have modified the following sentence to include a statement of our position: “While H2S donors remain a vital strategy to efficiently deliver this potent and protective molecules, we propose that exploiting the endogenous, cardiac H2S system can also be an integral and specific way of remediating heart failure and other cardiovascular diseases.” (Page 111; Line 386-389).
Minor comments
- Lines 64-65: not easy to understand the meaning of this sentence.
Author Response:We thank you for the clarification and have modified the sentence based on this comment and suggestions from Reviewer #1 (Page 2; Lines 62-66).
- Figure 1: This illustration employs conventional typology used to write chemical/enzymatic reactions but the intended meaning is unconventional. Due to this incoherence, the figure is not self-explained. A better and more elaborated scheme or illustration would help.
Author Response:We thank you for suggestions and have corrected Figure 1. We have removed the chemical conventions and illustrated the reactions to demonstrate the various sources of H2S production within a cell.
- Lines 206-207: this is an incorrect calculation: the canonical production of ATP from C18 stearic acid through beta oxidation and respiratory metabolism is 120 ATP/mol.
Author Response:We thank you for the clarification and have modified the sentence based on this comment and suggestions from Reviewer #1 (Page 2; Lines 62-66).
- Figure 2: This scheme attracts the attention excessively toward the electron transport chain and the whole process of mitochondrial respiration, which is shown in detail, and little is dedicated to the main point in this review: the fate of H2S. The lower part should be abbreviated or schematized and the “sulfite oxidation unit” should be expanded.
Author Response:We thank you for suggestions and have corrected Figure 2 so the electron transport chain is minimized and the sulfide oxidation unit is accented.
- Typographical errors have to be corrected: for instance, lines: 167, 170, 251, 321, 344, 404, 490, 502, 504, 505, 645.
Author Response: We thank you for noting these typographical errors and we have corrected them all.